# Evaluation of Virus-Free Chrysanthemum ‘Hangju’ Productivity and Response to Virus Reinfection in the Field: Molecular Insights into Virus–Host Interactions

**DOI:** 10.3390/plants13050732

**Published:** 2024-03-05

**Authors:** Xuejie Du, Xinqiao Zhan, Xueting Gu, Xinyi Liu, Bizeng Mao

**Affiliations:** 1Institute of Biotechnology, Ministry of Agriculture Key Lab of Molecular Biology of Crop Pathogens and Insects, Key Laboratory of Biology of Crop Pathogens and Insects of Zhejiang Province, Zhejiang University, Hangzhou 310058, China; xjdu@zju.edu.cn (X.D.); 0622730@zju.edu.cn (X.G.); 22316067@zju.edu.cn (X.L.); 2School of Pharmaceutical Sciences, Taizhou University, Taizhou 318000, China; joezhan@tzc.edu.cn; 3Zhejiang Tongxiang Hangbaiju Technology Academy, Tongxiang 314500, China

**Keywords:** *Chrysanthemum morifolium*, plant virus, ‘Hangju’, salicylic acid

## Abstract

The shoot apical meristem culture has been used widely to produce virus-free plantlets which have the advantages of strong disease resistance, high yield, and prosperous growth potential. However, this virus-free plant will be naturally reinfected in the field. The physiological and metabolic responses in the reinfected plant are still unknown. The flower of chrysanthemum ‘Hangju’ is a traditional medicine which is unique to China. In this study, we found that the virus-free ‘Hangju’ (VFH) was reinfected with chrysanthemum virus B/R in the field. However, the reinfected VFH (RVFH) exhibited an increased yield and medicinal components compared with virus-infected ‘Hangju’ (VIH). Comparative analysis of transcriptomes was performed to explore the molecular response mechanisms of the RVFH to CVB infection. A total of 6223 differentially expressed genes (DEGs) were identified in the RVFH vs. the VIH. KEGG enrichment and physiological analyses indicated that treatment with the virus-free technology significantly mitigated the plants’ lipid and galactose metabolic stress responses in the RVFH. Furthermore, GO enrichment showed that plant viral diseases affected salicylic acid (SA)-related processes in the RVFH. Specifically, we found that phenylalanine ammonia-lyase (PAL) genes played a major role in defense-related SA biosynthesis in ‘Hangju’. These findings provided new insights into the molecular mechanisms underlying plant virus–host interactions and have implications for developing strategies to improve plant resistance against viruses.

## 1. Introduction

*Chrysanthemum morifolium* ‘Hangju’ cultivar is native to Tongxiang, Zhejiang Province, China. The flowers of ‘Hangju’ can be processed into tea and are used to relieve pathogenic heat, protect the liver, improve eyesight, and aid detoxification [1,2]. Chrysanthemum tea is rich in a variety of secondary metabolites, such as flavonoids, alkaloids, and terpenoids, which are considered important active components in immune rehabilitation [2]. In chrysanthemum production, cutting is mainly used for seedling propagation, which can quickly expand and reproduce a large number of seedlings. However, this asexual propagation method also leads to the rapid spread of viruses, which can accumulate with the increase in planting years, resulting in poor growth and development of flowers and a decrease in yield and quality. Viruses in plants pose a significant threat to agricultural production, resulting in substantial costs for growers every year [3]. The analysis of four near-complete genome sequences representing the genetic diversity indicates that recombination might play a significant role in the evolution of the virus [4]. Up to now, all reported infections of chrysanthemum are RNA viruses [5], such as chrysanthemum virus B (CVB), chrysanthemum virus R (CVR), tomato aspermy virus (TAV), chrysanthemum stunt viroid (CSVd) [6,7,8]. CVB is the most common virus in ‘Hangju’, which can cause symptoms such as leaf mosaic, mottling, or vein clearing [5]. CVB is mainly transmitted by aphids and sap [4]. It is noteworthy that CVB, which is mainly prevalent in Zhejiang, China, belongs to the Carlavirus genus of the Betaflexiviridae family [8,9]. Producing and applying virus-free plants is an efficient method to reduce the losses caused by viral diseases. Various meristem tip culture methods are widely used to obtain virus-free plantlets [10,11,12]. The virus-free plants are widely planted worldwide, especially those of many vegetatively propagated economic crops such as agricultural, horticultural, and medicinal plants [13]. Eliminating plant viruses leads to higher yield, chlorophyll content, and enzyme activities, as well as a lower membrane lipid peroxidation degree [14]. Meanwhile, the removal of plant viruses can enhance plant stress resistance [15]. Nevertheless, the underlying mechanisms for the high yield and quality of virus-free plants are not fully understood.

To successfully survive under biotic stress conditions, plants have developed highly complex interactions among various phytohormones [16]. SA is a defense-related plant hormone that plays a key role in resistance to different microbial pathogens, such as viruses, bacteria, fungi, and oomycetes [17,18]. SA signaling constitutes the major defensive pathway against viruses [19]. The recognition of viral effectors by R proteins triggers defensive mechanisms, including activation of the SA and siRNA pathways, ROS production, and the hypersensitive response (HR) [20]. Auxin opposes the SA pathway, and specific auxin response factors (ARFs) are crucial for the replication and spread of certain viruses, such as tobacco mosaic virus (TMV) [21,22]. Ethylene (ET) also counteracts the pathway downstream of SA signaling, and plays a role in the development of symptoms caused by cauliflower mosaic virus infection, the systemic movement of TMV, and the formation of necrotic lesions following infection by other viruses [23,24]. Jasmonic acid (JA) supports plant defense at early stages of infection, but, if it is induced or applied at later stages, it decreases plant resistance [25,26]. Abscisic acid (ABA) plays complex roles in plant defense: it enhances callose deposition on plasmodesmata (PD) to limit virus movement between cells, but also opposes the SA pathway, reducing local resistance by suppressing HR induction, decreasing ROS and SA production, and weakening distal systemic acquired resistance (SAR) and siRNA systems [27,28,29]. The SA, JA, and ET signaling pathways interact extensively [30]. Their regulatory cross-talk may have evolved to allow plants to fine-tune the induction of their defenses in response to different plant pathogens [17]. Thus, the mediation of broad-spectrum antiviral immunity by phytohormones is important in plants.

We previously built a virus-free system in ‘Hangju’ and increased the soil bacterial diversity and root productivity in the field [13]. However, the mechanism behind the reinfection of virus-free seedlings in the field remained unknown. In this study, we determined the transcriptomes of RVFH and VIH and provided new insight into the mechanism of RVFH’s response to virus reinfection in the field.

## 2. Results

### 2.1. Virus Detection in RVFH and VIH

A VFH tissue culture system was established in our previous study [7]. The virus elimination significantly improves the root growth and overall yield of ‘Hangju’ [13]. However, it is unclear whether the VFH is reinfected with viruses in the field. During a survey of viral diseases in the field in September 2021, VFH leaves had no clear leaf mottling and vein yellowing compared with VIH (Figure 1A). The chlorophyll contents of the VFH significantly increased 1.2-fold compared with VIH (Figure 1B). We previously reported that detectable viruses of VIH are CVB [13]. The reverse transcription polymerase chain reaction (RT-PCR) results showed the mature stage of VFH leaves reinfected with CVB in the field (Figure 1C). Consequently, we rename the reinfected VFH in the field as RVFH. Furthermore, we generated six whole-transcriptome libraries derived from flowers, with three biological replicates from RVFH and three biological replicates from VIH. After quality control, the sequences were mapped against the chrysanthemum genome to remove sequences derived from the ‘Hangju’ host. Then, the sequences of RVFH libraries and VIH libraries were separately assembled de novo. The assembled contigs were blasted against the NCBI reference viral genome database with a cutoff E-value of 1 × 10^−10^ and a 90% match ratio. Ten viruses were identified in VIH, including six contigs of CVB and forty-one contigs of CVR (Figure 1D). Eleven viruses were identified in RVFH, including 80 contigs of CVB and 313 contigs of CVR (Figure 1E). The unmatched sequences were separately mapped into the CVB and CVR genome, producing seven virus genes with expression. These gene expression levels all significantly decreased in RVFH vs. VIH (Figure 1F). The semi-quantitative RT-PCR results demonstrate that *TGBp2* expression levels of CVB and CVR significantly decreased in RVFH vs. VIH (Figure 1G). However, 10 viruses reported to infect chrysanthemums were not identified using RT-PCR (Appendix A). Thus, RVFH was infected with CVB and CVR in the field and represses virus accumulation at the transcript level.

### 2.2. Effect of Natural Infection on RVFH Growth

The growth phenotype of the RVFH was significantly different from that of the VIH in the field (Figure 2A). The flowers of RVFH were bigger than that of VIH (Figure 2B). RVFH exhibited better development than VIH. The plant height, stem diameter, and branching number significantly increased in RVFH plants by 1.8-fold, 1.3-fold, and 2.3-fold, respectively, compared with VIH plants (Figure 2C). Similarly, flower growth, including flower diameter, effective bud count, and flower yield, significantly increased in RVFH plants by 1.3-fold, 2.3-fold, and 3.5-fold, respectively, compared with VIH plants (Figure 2D). Chlorogenic acid, luteolin, and 3,5-dicaffeoylquinic acid levels were important active ingredients to reflect the quality of the chrysanthemum. They also significantly increased by 1.4-fold, 1.3-fold, and 1.4-fold in RVFH vs. VIH, respectively (Figure 2E).

The cross-section (Figure 2F) and vertical section (Figure 2H) showed growth of the stem during the vegetative growth period of the chrysanthemum. The cross-sectional area and longitudinal diameter (Figure 2G,I) of the stem in RVFH were significantly higher than those in VIH, with a 1.6-fold and 1.2-fold, increase, respectively. These observations indicated that virus-free technology was important for plant growth, flower yield, and quality in ‘Hangju’.

### 2.3. KEGG Enrichment Analysis of DEGs in RVFH vs. VIH

We collected flower tissues from RVFH and VIH for transcriptomic sequencing (three biological replicates from RVFH and three biological replicates from VIH). The raw reads were qualified, and adapters were removed, generating 224,752,165 clean reads from VIH libraries and 202,909,472 clean reads from RVFH libraries (Appendix A). To review a summary of the expression matrix, a principal components analysis (PCA) was performed. The PCA results for RVFH and VIH were separated into two groups, and the reductive values of PC1 and PC2 were 93.85% and 3.63%, respectively (Figure 3A). A fold change (FC) > 1 and *p* (adjusted value) < 0.01 were used as cutoffs to identify differentially expressed genes (DEGs). In total, 1280 genes were upregulated and 4943 genes were downregulated in RVFH vs. VIH, respectively (Figure 3B). KEGG enrichment analysis revealed that three pathways were significantly enriched (*p* adjusted value < 0.05) in RVFH vs. VIH, including ‘Sphingolipid metabolism’, ‘Fatty acid elongation’, and ‘Galactose metabolism’ (Figure 3C). We further identified the numbers of genes related to these three pathways, and most of them were downregulated in RVFH vs. VIH. However, we found that two genes, *scaffold28G000114* and *Cm14G001475*, increased 2.5-fold in RVFH vs. VIH (Figure 3D). *Scaffold28G000114* encodes an alpha-galactosidase which is involved in galactosylceramide synthesis. *Cm14G001475* encodes a glucosylceramidase which is involved in ceramide synthesis (Appendix A). In ‘Fatty acid elongation’, the genes that encoded for 3-ketoacyl-CoA synthases exhibited lower expression levels in RVFH vs. VIH, except for *Cm25G001904* which increased 3.3-fold in RVFH vs. VIH (Figure 3E). In ‘Galactose metabolism’, 37 genes showed significant differences, namely 10 increased and 27 decreased in RVFH vs. VIH (Figure 3F). These results suggested that the RVFH had significantly alleviated lipid and galactose metabolic stress responses during virus infection.

### 2.4. GO Enrichment Analysis of DEGs in RVFH vs. VIH Plants

In our study, 129 GO terms were significantly enriched (adjusted *p* value < 0.01) in the biological process (BP) category in RVFH vs. VIH (Figure 4A and Appendix A). Moreover, thirty GO terms and four cell wall-related GO terms were significantly enriched (adjusted *p* value < 0.01) in the molecular functions (MF) category and cellular components (CC) category, respectively (Figure 4A and Appendix A). Five phytohormone-related and superoxide-related pathways were selected from BP, and then the selected DEGs were clustered into two groups for comparison (Figure 4B). Four genes expression levels increased in RVFH vs. VIH, including a 3.1-fold increase in *Cm06G004022* (ABC transporter), a 2.7-fold increase in *Cm14G003264* (indole-3-pyruvate monooxygenase), a 2.5-fold increase in *Cm13G001188* (zinc finger), and a 2.5-fold increase in *Cm27G003733* (WD40). However, most DEGs were downregulated in RVFH vs. VIH, including SA-responsive genes (Figure 4B). Furthermore, we detected some physiological parameters related to antioxidant activity, including ROS activity, malondialdehyde (MDA) content, superoxide dismutase (SOD), peroxidase (POD), and catalase (CAT) activities (Figure 4C). These indexes were significantly higher in VIH than those in RVFH, indicating that the level of ROS in plants infected with the virus increased, the degree of cell damage deepened, and the activities of various antioxidant enzymes increased accordingly.

### 2.5. Regulation of Salicylic Acid Associated with Plant Virus Elimination in RVFH vs. VIH

SA is produced from chorismate via two independent pathways in plants, including isochorismate synthase (ICS) and phenylalanine ammonia-lyase (PAL) [31,32]. Transcriptomic analysis revealed that the ‘phenylalanine ammonia-lyase activity’ term of MF was significantly enriched (*p* adjusted value = 1.15 × 10^−5^) in RVFH vs. VIH, including eleven *PALs* (Figure 4A and Appendix A). Furthermore, four putative *ICS* genes and twenty-eight *PAL* genes were identified from the *C*. *morifolium* genome. The expression patterns of all *ICS* and *PAL* genes were investigated and almost all were downregulated in RVFH vs. VIH (Figure 5A and Appendix A). Phylogenetic analysis showed that three *ICS* of *C*. *morifolium* and one *ICS* of *C*. *seticuspe* were classified into one clade. *Cm23G000276* was separated from other chrysanthemum *ICS* (Figure 5B). No expression levels of *Cm23G000276* were detected in *C*. *morifolium* flowers (Figure 5A). A quantitative real-time RT-PCR (RT-qPCR) was performed and three *ICS* and five *PALs* showed similar expression patterns in their transcriptome data (Appendix A). These results suggested that *ICS* from *C*. *morifolium* might play a minor role in SA-induced synthesis. Based on the phylogenetic analysis of *PALs* from chrysanthemum, rice, and *Arabidopsis*, 48 *PALs* were divided into six groups, including a core *PLAs* clade ‘Group a’ (Figure 5C). Similarly, multiple copies of *PALs* in *C*. *morifolium* were distributed among the other five groups. These results suggested that the *PALs* family was probably associated with polyploidization events in chrysanthemum evolution. Collinearity analysis was performed on Twenty-eight *PALs* from *C*. *morifolium* and six *PALs* from *C*. *seticuspe*. Three pairs of orthologous gene pairs were found between *C*. *morifolium* and *C*. *seticuspe* (Figure 5D). Co-expression analysis showed there was a positive correlation between PLAs and the SA-response pathway in eight of the pairs (Figure 5E). Moreover, SA levels were significantly increased in VIH compared with RVFH (Figure 5F). PAL catalyzed the chorismate-derived L-phenylalanine into cinnamic acid, which also provided precursors to the flavonoid pathway. Total flavonoid levels were significantly higher in VIH than in RVFH (Figure 5G). However, no flavonoid metabolic-related pathways were enriched through KEGG and GO (Appendix A). These results suggested that *PALs* played a major role in defense-related SA biosynthesis in *C*. *morifolium*.

## 3. Discussion

Plant viruses can have a significant impact on the growth and biomass accumulation of plants. Studies have demonstrated that healthy virus-free seedlings derived from plant tissue culture show specific advantages over virus-infected seedlings in terms of agronomic, physiological, and biochemical indicators, as well as yield and quality. These advantages may include higher growth rates, healthier root systems, enhanced photosynthetic efficiency, and improved quality [13,14]. Nevertheless, the studies merely concentrated on plant virus detection during the tissue culture phase. Studies on the reinfection of virus-free seedlings in the field are still lacking. In this study, we investigated the occurrence of virus infections in VFH under natural conditions. Though VHF leaves have no symptoms, CVB was detected in the VFH (Figure 1C). The transcriptome and RT-PCR analysis revealed the expression levels of CVB and CVR significantly decreased in RVFH compared with VIH (Figure 1F,G). We speculated that RVFH still maintains immunity against virus infection by suppressing the expression of virus genes. Viral infections affect the physiological activities of plants, and resistance in plants is directly related to the degree of damage. Many studies assume that ROS production is a basic symptom of plant toxicity under stress [33,34]. As the production of ROS increases, the phytotoxicity rises too [35]. MDA, a product of peroxidation, has a toxic effect on cells, and its content can reflect the level of membrane lipid peroxidation. In such toxic conditions, plant growth and metabolism are adversely affected, resulting in lower crop productivity [35]. Our results revealed that VIH accumulates more ROS and MDA than RVFH, resulting in a lower biomass (Figure 2). In plants, ROS causes serious damage to the cells by inhibiting proteins, DNA, and other metabolic pathways. Moreover, the defense system against ROS is activated in the plants to regulate its functional activity by activating different enzymatic and non-enzymatic antioxidant agents [35]. The activities of SOD, POD, and CAT were significantly higher in RVFH vs. VIH (Figure 4C). In addition, the non-enzymatic defense system [36] also responded, as indicated by a significantly higher flavonoid content in VIH than in RVFH (Figure 5G). However, ROS production and scavenging still cannot be balanced, so the final result was a decrease in yield and quality.

The present study provided valuable insights into RVFH response to lipid and galactose metabolism, as well as SA-related processes in *C*. *morifolium*. The downregulation of genes involved in ‘Sphingolipid metabolism’, ‘Fatty acid elongation’, and ‘Galactose metabolism’ pathways in RVFH vs. VIH plants suggests that plant virus infection imposes significant metabolic stress on the host plant (Figure 2). This is consistent with previous studies showing that viruses can manipulate the host’s metabolism to enhance their replication and spread [37]. However, the mechanism underlying the regulation of lipid metabolism in RVFH is largely unknown. Viral proteins interacted with peroxisomal proteins, such as HIV’s Nef and influenza’s NS1, or used the peroxisomal membrane for RNA replication [38]. Peroxisomes are sites of lipid biosynthesis and catabolism, reactive oxygen metabolism, and other metabolic pathways [38]. Based on GO enrichment analysis, ‘response to ozone’ was significantly enriched in RVFH vs. VIH (Figure 4A). This pathway contains genes that were all downregulated in RVFH vs. VIH, including superoxide dismutase, lipoxygenase, and long-chain acyl-CoA synthetase (Figure 4B). ROS content and ROS-related enzyme activity significantly decreased in RVFH vs. VIH (Figure 4C). ROS increases are closely related to total lipids under stress conditions [39]. Peroxisomes are important organelles in plant cells that participate in various physiological and developmental processes, such as fatty acid β-oxidation and the biosynthesis of hormones and signal molecules [40]. Thus, plant virus reinfection led to extensive remodeling of lipid metabolic pathways in RVFH.

The clearest role of SA is to regulate the response of plant defense mechanisms to pathogen infection. SA is synthesized in plants by the ICS pathway and the PAL pathway. The *Arabidopsis* genome contains two ICS genes, ICS1 plays a major role, whereas ICS2 plays a minor role, in pathogen-induced SA synthesis [32,41]. Only one ICS was detected in the rice genome (Figure 5B). Basal SA levels are very low in most plants. However, rice contains a higher basal level of SA under normal conditions. SA levels in *Arabidopsis* do not greatly increase after pathogen infection [42]. These findings reveal that the regulation of SA biosynthesis varied significantly between species. Our results also demonstrated that plant virus reinfection affected SA signaling and accumulation in RVFH (Figure 4 and Figure 5). The downregulation of most SA-responsive genes in RVFH vs. VIH plants suggested that RVFH might suppress SA-mediated defense responses (Figure 4B).

Recently, a peroxisome beta-oxidation enzyme (AMT1) was found to participate in the regulation of the PAL pathway in rice [43]. Interestingly, our study also revealed a positive correlation between PAL genes and the SA-response pathway through co-expression analysis (Figure 5). This suggested that *PAL* genes might be transcriptionally activated by SA signaling to enhance SA synthesis during plant virus infection. Moreover, our results showed that total flavonoid levels were significantly higher in VIH than in RVFH (Figure 5G). Flavonoids have been implicated in various physiological processes, including defense responses [44,45,46]. However, no flavonoid-related pathways were enriched through KEGG and GO analyses in our study. Therefore, further investigation is needed to determine the relationship between flavonoids and virus reinfection in RVFH.

## 4. Materials and Methods

### 4.1. Plant Materials and Sampling

The leaves of VIH in the field had been investigated in our previous study [13]. The VIH seedlings were stored and propagated from cuttings in the greenhouse of Tongxiang (N: 30°38′, E: 120°32′), Zhejiang, China. The shoots’ meristem tips were excised to remove viruses, and the detailed method has been described in our previous study [7,13]. The 30-day-old VFH seedlings and VIH seedlings were separately transplanted to two adjacent fields in March 2021 in Tongxiang. During a field survey in September 2021, we collected VFH leaves and performed detection for virus infection. Samples of RVFH and VIH were collected from five locations in the field, with distances ranging from 50 to 100 m apart. For physiological index analysis, the apical third leaves and top flowers of individual plants were collected from five locations. For transcriptome analysis, flowers of three individual plants were separately collected from three locations. Each sample was collected in a sterile sealed bag and then frozen immediately in liquid nitrogen and stored at −80 °C for RNA extraction and other detection methods.

### 4.2. Microscopic Observation of Paraffin Section of Stem

The microstructure of the stems was observed by toluidine blue staining according to Sakai’s method [47]. At the vegetative stage, stems below the apical growth point of RVFH and VIH plants were collected and sent to Hangzhou Haoke Biotechnology Company (Hangzhou, China) for paraffin section staining and imaging.

### 4.3. Determination of Reactive Oxygen Species Levels and Antioxidant Capacity

ROS was detected using the tetramethylbenzidine (TMB) chromogenic method [48]. ROS was detected using an ELISA kit (SINOBESTBIO, Beijing, China). A microplate reader (Flexstation 3, Molecular Devices, San Jose, CA, USA) was used to measure the absorbance at 450 nm and the sample’s activity was calculated. MDA was detected using the thiobarbituric acid (TBA) method [49]. The MDA in lipid peroxide degradation products can be condensed with TBA, forming red products with a maximum absorption peak at 532 nm. MDA content was measured using test kits (Nanjing Jiancheng Bioengineering Institute of China, Nanjing, China) following the protocols described by the manufacturers. For protein activity detection, the tissues (0.1 g per piece) were cut into pieces and then ground in a grinder. The grinding instrument (Jingxin, China) was precooled at 60 Hz for 10 s at an interval of 20 s 3 times to prepare 10% tissues homogenate and then centrifuged (Centrifuge 5427R, Hamburg, Germany) at 8000 r/min at 4 °C for 10 min before taking the supernatant for determination. The activity of SOD was detected by the WST-1 method [50]. The activity of POD was detected by the guaiacol chromogenic method [51]. The activity of CAT was detected according to the method described by Guo [52].

### 4.4. RNA Extraction, Library Construction, and Sequencing

Total RNA was extracted using the RNA Kit (TIANGEN, Beijing, China). The methods of RNA quantity, cDNA library preparation, and transcriptomic analysis were the same as in our previously published work [53]. In brief, expression levels for each unigene were calculated as the FPKM using an in-house script. The DEGs were screened based on the criterion: FDR ≤ 0.05, log_2_fold-change (FC) > 1 or log_2_FC < −1, and with statistical significance (*p* adjusted-value < 0.05). The DEGs were also subjected to GO enrichment analysis and KEGG pathway enrichment analysis using ClusterProfiler (v4.2.2) [54]. To identify viruses in the chrysanthemum transcriptomes, the no-match reads were assembled de novo using the Trinity program (https://github.com/trinityrnaseq/trinityrnaseq, accessed on 24 February 2024). The contigs assembled from transcriptome were blasted against NCBI’s reference viral genome database (http://www.ncbi.nlm.nih.gov/genome/viruses/, accessed on 24 February 2024). In addition, the no-match reads were mapped into CVB and CVR genomes (OQ335844.1, NC_040703.1, https://www.ncbi.nlm.nih.gov/datasets/genome/, accessed on 24 February 2024) to analyze virus genes’ expression. The RNA-seq data have been submitted to the BIG Data Center of the Chinese Academy of Sciences (http://bigd.big.ac.cn, accessed on 24 February 2024) with accession number CRA014394.

### 4.5. Determination of SA and Total Flavonoid Content

Using ammonium iron sulfate as the chromogenic agent, the purplish-red complex formed by SA and Fe^3+^ has the maximum absorption at 472 nm. Briefly, a standard curve was first made using the SA standard (Sinopharm Group Chemical Reagent Co., Ltd., Shanghai, China). The flowers were then ground into a powder, and 0.5 g of the powder was accurately weighed in 10 mL absolute ethanol for ultrasonic extraction for 30 min. Then, the absorbance was measured at 472 nm using ammonium iron sulfate (Shanghai YI EN Chemical Technology Co., Ltd., Shanghai, China) as a chromogenizer. Finally, the corresponding SA concentration was calculated from the standard curve and then converted to a mass ratio format.

Total flavonoids were detected using the NaNO_2_-Al(NO_3_)_3_-NaOH chromogenic method. A standard curve was first made using rutin standards (Solarbio, Beijing, China). The dried samples were ground into powder, weighed out to 1 g, added to 30 mL 70% ethanol, soaked for 24 h, and then filtered. The filtrate was treated with 5% NaNO_2_ (Sinopharm Group Chemical Reagent Co., Ltd., Shanghai, China), 10% Al(NO_3_)_3_ (Shanghai Aladdin Biochemical Technology Co., Ltd., Shanghai, China), and 4% NaOH (Sinopharm Group Chemical Reagent Co., Ltd., Shanghai, China). The absorbance was measured at 510 nm, and the total flavonoid concentration was calculated according to the standard curve, and finally converted to mass ratio.

### 4.6. Determination of Pharmacodynamic Components

The content of pharmacodynamic components is a necessary indicator of the quality of ‘Hangju’. According to Chinese Pharmacopoeia, the contents of chlorogenic acid, luteolin, and 3, 5-O-dicafeoylquinic acid in chrysanthemum should be at least 0.20%, 0.08%, and 0.70%, respectively. The three components use the procedures described in the Chinese Pharmacopoeia (version 2020).

### 4.7. RT-PCR and RT-qPCR

Total RNA was isolated from different samples using the TransZol reagent (TransGen Biotech, Beijing, China). RNA extracts were treated with DNaseI (NEB, Hitchin, UK) to eliminate DNA contamination. First-strand cDNA was produced from the RNA template by reverse transcription using the TIANscriptRTKit according to the manufacturer’s instructions (TransGen Biotech, Beijing, China). The RT-PCR analyses were performed as described previously [13]. The semi-quantitative RT-PCR and RT-qPCR analyses were performed as described previously [45]. The primers used for real-time PCR are listed in Appendix A.

### 4.8. Statistical Analysis

The data are displayed as means ± standard deviations (SD). Statistical analysis was performed using the GraphPad Prism software (version 9) [55] and assessed using a one-way analysis of variance and the least significant difference was calculated at *p* ≤ 0.05 using Student’s *t*-test. Data were also treated by hierarchical clustering with the R package pheatmap (v.1.0.12, accessed on 7 July 2023) and by Principal Component Analysis using the R package FactoMineR (v.1.2, accessed on 7 July 2023).

## 5. Conclusions

Our study provided new insights into the molecular mechanisms underlying plant virus–host interactions and had implications for developing strategies to improve plant resistance against viruses. The results of this study suggested that RVFH alleviates lipid and galactose metabolic stress responses in chrysanthemum and affects SA-related processes by modulating the expression of PAL genes involved in defense-related SA biosynthesis. Further research is needed to fully understand the complex network of interactions between viruses and RVFH and to develop effective strategies for controlling viral plant diseases.

## Figures and Tables

**Figure 1 plants-13-00732-f001:**
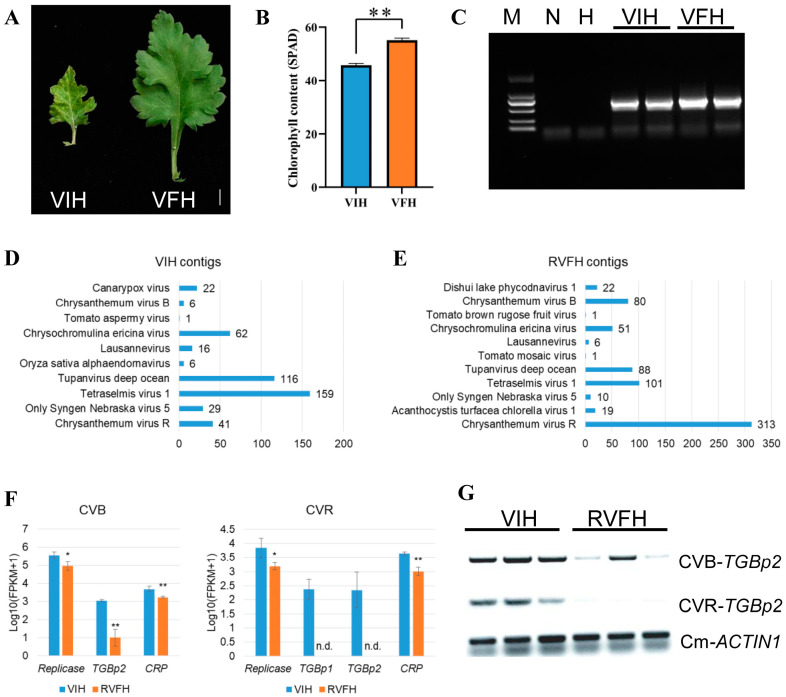
Detection of plant viruses in the field. (**A**) Leaf symptoms of VIH and VFH in the field. Scale bar = 0.5 cm. (**B**) Chlorophyll content of VIH and VFH (n = 5). (**C**) CVB and CVR were detected by RT-PCR (35 cycles). M, mark; N, ddH_2_O; H, the healthy virus-free seedling in tissue culture. (**D**) The number of contigs in VIH. (**E**) The number of contigs in RVFH. Selected with an E-value of 1 × 10^−10^ and a 90% match ratio of virus sequences in NCBI viral reference genome database. (**F**) The expression levels of CVB and CVR genes (n = 3). TGBp, triple gene block protein; CRP, cysteine-rich protein; n.d., no detection. (**G**) The expression levels of *TGBp2* were detected by semi-quantitative RT-PCR (28 cycles). *C*. *morifolium ACTIN* (*CmACTIN*) was used as an internal control. Values are the mean ± SD. Student’s *t* test; * *p* < 0.05; ** *p* < 0.01. The primers are listed in Appendix A.

**Figure 2 plants-13-00732-f002:**
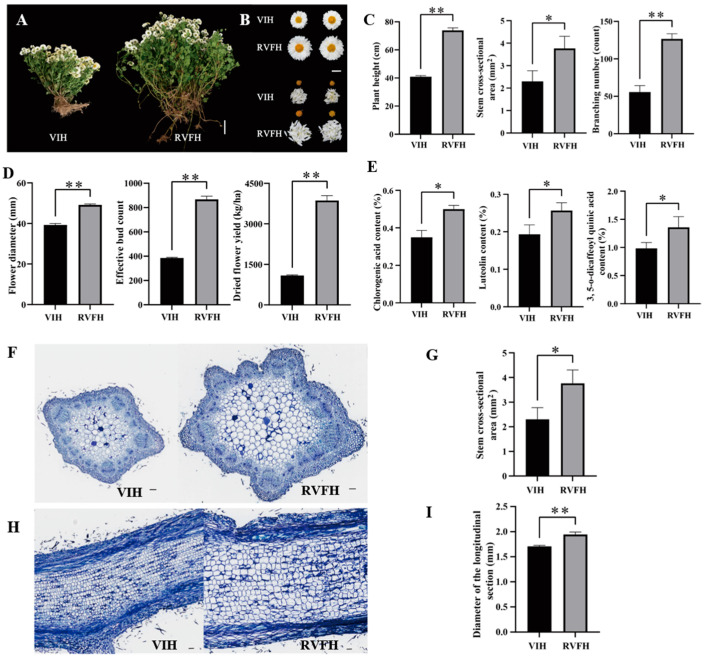
RVFH physiological responses in the field. (**A**) Plant phenotypes of ‘Hangju’ of VIH and RVFH. Bar = 10 cm. (**B**) Flower phenotypes of ‘Hangju’ of VIH and RVFH. Bar = 2 cm. (**C**) Growth indicators of VIH and RVFH (n = 10). (**D**) Growth indicators of flower (n = 20). (**E**) Quality-related medicine components of VIH and RVFH (n = 3). (**F**) Stem cross-section of VIH and RVFH. Bar = 0.1 mm. (**G**) Stem cross-sectional area of VIH and RVFH (n = 50). (**H**) Stem longitudinal section of VIH and RVFH. Bar = 0.1 mm. (**I**) Diameter of the longitudinal section of VIH and RVFH (n = 50). Values are the mean ± SD. Student’s *t* test; * *p* < 0.05; ** *p* < 0.01.

**Figure 3 plants-13-00732-f003:**
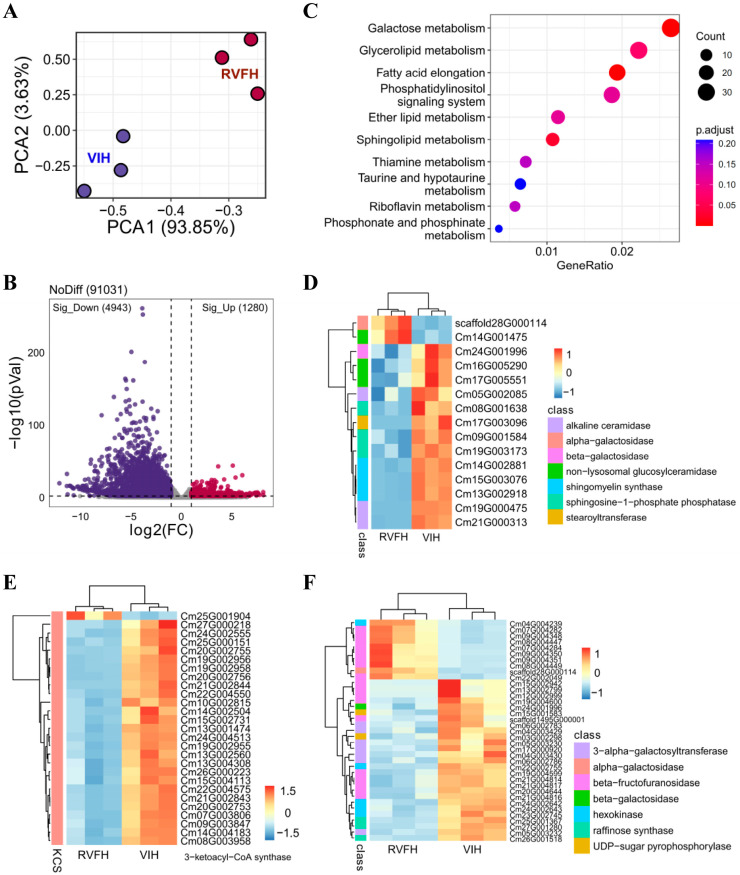
KEGG enrichment analysis of DEGs in RVFH vs. VIH plants. (**A**) PCA analysis of the transcriptome. (**B**) Volcano plot showing DEGs in RVFH vs. VIH plants. (**C**) Top ten enriched pathways in RVFH vs. VIH plants. The color bar represents the levels of adjusted *p* value. (**D**) Expression profiles of sphingolipid metabolism-related genes RVFH vs. VIH plants. (**E**) Expression profiles of fatty acid elongation-related genes RVFH vs. VIH plants. (**F**) Expression profiles of galactose metabolism-related genes RVFH vs. VIH plants. The color bar represents the normalization for log_2_-FPKM using the Pheatmap software package (v1.0.12).

**Figure 4 plants-13-00732-f004:**
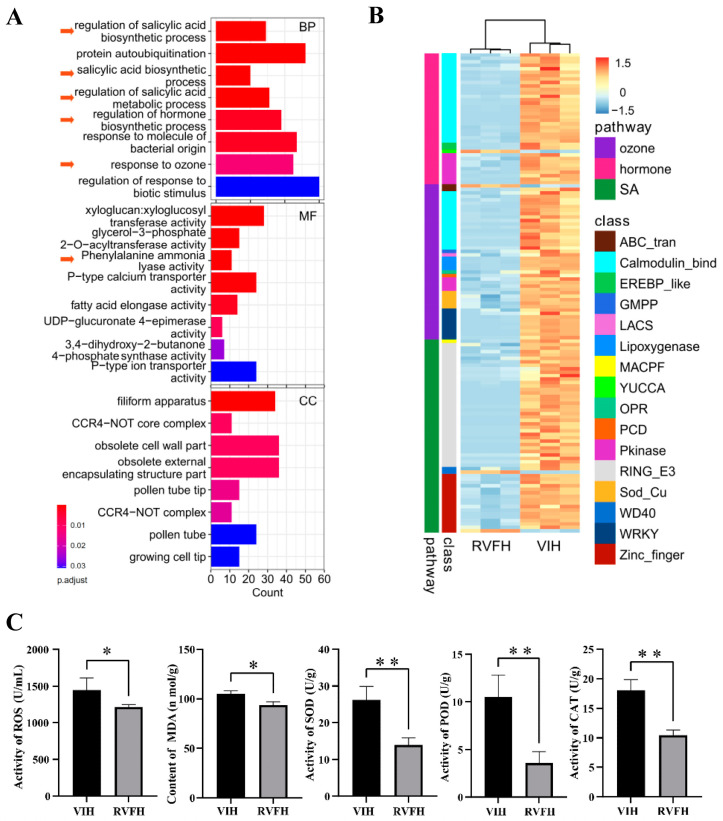
GO enrichment analysis of DEGs in RVFH vs. VIH. (**A**) Top eight pathways in each GO category. The color bar represents the levels of *p* value. BP, biological process; MF, molecular functions; CC, cellular components. The pathways related to phytohormones and superoxide were labeled with arrows. (**B**) Expression profiles of ozone-, hormone-, and SA-related pathway in RVFH vs. VIH. The color bar represents the normalization for log_2_-FPKM using the Pheatmap software package (v1.0.12). (**C**) The levels of ROS, MDA, SOD, POD, and CAT in RVFH and VIH (n = 5). Values are the mean ± SD. Student’s *t* test; * *p* < 0.05; ** *p* < 0.01.

**Figure 5 plants-13-00732-f005:**
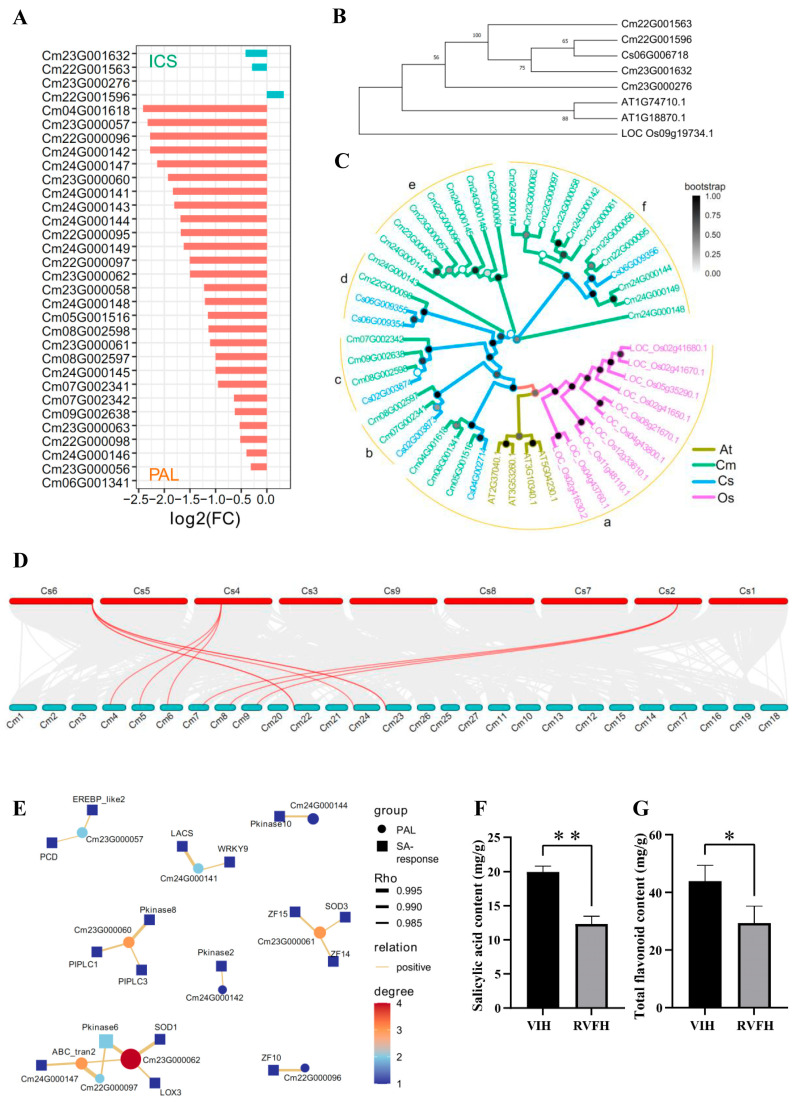
Integrated analysis of SA regulation in RVFH vs. VIH. (**A**) Expression profiles of *ICSs* and *PALs*. (**B**) Phylogenetic analysis of ICS in chrysanthemum, rice and *Arabidopsis*. A total of 8 ICSs were used to construct the unrooted maximum-likelihood phylogenies. (**C**) Phylogenetic analysis of PAL in chrysanthemum, rice, and *Arabidopsis*. A total of 48 PALs were used to construct the unrooted maximum-likelihood phylogenies. The 48 PALs were divided into six groups, each labeled with a letter. At, Arabidopsis; Cm, C. morifolium; Cs, C. seticuspe; Os, Oryza sativa. (**D**) Collinear analysis for the PALs between *C*. *morifolium* and *C*. *seticuspe*. (**E**) Co-expression analysis of *PALs* and SA-related genes in RVFH vs. VIH. The solid lines represent positive correlations. The thickness of the line is determined by a Pearson correlation coefficient > 0.9. The colors represent the correlated gene number. SA (**F**) and total flavonoid (**G**) contents in RVFH and VIH (n = 5). Values are the mean ± SD. Student’s *t* test; * *p* < 0.05; ** *p* < 0.01.

## Data Availability

RNA sequencing data produced in present study have been deposited in CRA014394.

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
