# Peer review of "Evaluation of Virus-Free Chrysanthemum ‘Hangju’ Productivity and Response to Virus Reinfection in the Field: Molecular Insights into Virus–Host Interactions"

_plants, 2024, doi:10.3390/plants13050732_

Round 1

Reviewer 1 Report

Comments and Suggestions for Authors

The manuscript gives new information about effect of plant virus elimination on physiological and metabolic pathways in chrysanthemum. But I think the manuscript has some problems and it requires “minor revisions” for publication.

1. virus-carrying plants

Authors discuss difference between VF plants and VC plants in various phenotypes and gene expression. But they do not describe which viruses were infected with VC plants. Each virus has specific symptoms. There are difference of phenotypes and gene expression among infected viruses.

2. transcriptomic sequences

In transcriptomic analysis, authors use only flower tissues. They should discuss why transcriptomic analysis is performed using only flower tissues.

3. figure

In all figures, resolution of a character image is low.

Figure explanation is insufficient.

Reviewer 2 Report

Comments and Suggestions for Authors

This article by Du et al., is not written well. I will highly suggest, please follow the journal instruction for writing MS (https://www.mdpi.com/journal/plants/instructions#submission). Page number is missing. Although the study looks interesting, but authors does not justify the results with desired experiments. In my opinion, the present MS is not ready for publication.

Major Comment;

Page no. 1; Abstract section; “Our previous study provided new insights” …. It is not a good idea to start your abstract with our previous study ……...!!  General readers might find difficulties in understanding this way.

Page no. 1; Abstract section; “virus-carrying” It is not a suitable terminology, meaning might be confusing too.

Page no. 1; Abstract section; salicylic acid-related (SA) process….” In next line it can be abbreviated.

Page no. 1; Introduction section; “cause plant crop losses annually” Are you discussing about C. morifolium ‘Hangju plant yield loss? It is not clear with this sentence.

Page no. 1; Introduction section; “more than 20 viruses” Are they 20 different species, DNA virus, RNBA virus? What are they?

Page no. 2; Introduction section; “It is mainly transmitted by aphid species (which one? And can be mechanically transmitted using sap”

Page no. 2; Introduction section; “Salicylic acid (SA)”

Page no. 2; Introduction section; “weakening distal SAR” do not abbreviate words if using first time in the MS. Follow the same throughout writing.

Page no. 2; Introduction section; “phytohormones in VF plants” What is VF plants? Several studies has been done on phytohormones role, function, characterization in virus infected plants (https://doi.org/10.1093/jxb/erab061, and https://doi.org/10.1111/nph.17261). I will suggest author’s to be very specific while writing these sentences.

Page no. 2; Result section; Figure 1A “seen that VF plants have larger biomass” Why the authors have not accessed individual plants? What is the difference if you take individual virus-free plants vs. virus infected plants. How many flowers did you assessed? And how you infected these plants?  Did you confirm the absence of any other virus in these plants?

Page no. 3; Result section; “virus elimination had a strong effect on the stem development” This could be due to nutritional effect? Justify!

Page no. 4; Result section; How many samples the authors have collected for transcriptomic sequencing, mention clearly.

Page no. 4; Result section; “The raw reads were qualified, and adapters were removed, generating 131.14 Gb of clean read, including 63.76 Gb from VC plants and 67.38 Gb from VF plants” No need to mention these much Gb of reads were clean! You can mention the number of reads. Or the authors should add map of reads observed.

Page no. 6; Result section; “Furthermore, we detected some physiological parameters related to antioxidant activity, including ROS activity, malondialdehyde (MDA) content, superoxide dismutase (SOD), peroxidase (POD), and catalase (CAT) activities (Figure 4C)”

Page no. 7; Result section; Arabidopsis

Page no. 7; Result section; “No expression levels of Cm23G000276 were detected in C. morifolium flowers” Di the authors have validated the gene expression using Real-time PCR? Some of the pathways genes need to be tested for confirmation of these results.

Comments on the Quality of English Language

Moderate editing is required. 

Round 2

Reviewer 2 Report

Comments and Suggestions for Authors

Authors have significantly improved the MS. However, some minor corrections needed, such as;

1.  "No sequences matching the chrysanthemum genome are extracted from three RVFH libraries and three VIH libraries" Re-write the sentence. 

Comments on the Quality of English Language

Minor typo error.
